# Fasys: Visible-Light-Based Communication and Positioning Services towards Smart Cities

**DOI:** 10.3390/s23146340

**Published:** 2023-07-12

**Authors:** Baozhu Yu, Xiangyu Liu, Lei Guo, Xuetao Wei, Song Song

**Affiliations:** 1School of Information Science and Engineering, Shenyang Ligong University, Shenyang 110158, China; yubz19@mails.jlu.edu.cn; 2Department of Computer Science and Engineering, Southern University of Science and Technology, Shenzhen 518055, China; weixt@sustech.edu.cn; 3School of Communication and Information Engineering, Chongqing University of Posts and Telecommunications, Chongqing 400065, China; guolei@cqupt.edu.cn (L.G.); songsong_eric@163.com (S.S.); 4Hangzhou Institute of Advanced Technology, Hangzhou 310056, China; 5Institute of Intelligent Communications and Network Security, Chongqing University of Posts and Telecommunications, Chongqing 400065, China

**Keywords:** visible light communication, visible light positioning, sensor data, smart cities

## Abstract

Visible-light-based transmission application plays an important role in various types of sensor services for the Internet of Things (IoTs). However, in big data scenarios, current visible-light-based systems cannot achieve concurrent high-speed communication, low-speed communication, and positioning. Therefore, in this article, we propose a smart visible-light-based fusion applications system, named Fasys, to solve the above problem for the big data traffic with heterogeneity. Specifically, for low-speed data services, we propose a novel linear block coding and bit interleaving mechanism, which enhances the LED positioning accuracy and recovers the lost data bits in the interframe gap (IFG). For high-speed data services with traffic possessing burstiness, an elegant statistical reliability analysis framework in regard to latency is proposed based on martingale theory. The backlog martingale process is constructed. Leveraging stopping time theory, a tight upper bound of unreliability is obtained. An arrival abstraction and traffic allocation scheme is designed, which contributes to decouple the reliability requirement as the maximum supportable arrival load. Finally, we implement our Fasys system, and extensive experimental results show that our system can achieve consistent high-precision positioning and low-BER data communication for low-speed data services. And the proposed martingale-based traffic allocation scheme can achieve the provisioning of reliability in regard to the latency for high-speed data services.

## 1. Introduction

Both smart cities and smart homes derived from the Internet of Things (IoTs) are popularized in people’s lives. Modern people are willing to spend 80% to 90% of their daily time in indoor spaces for activities [1,2], which produce multi-modal data, e.g., GPS data, image data, video data, and audio data. High-speed Internet surfing and colorful machine-to-machine interconnections (low-speed) have become rich application requirements. The networks are expected to embrace the massive data traffic with heterogeneous characteristics. For example, you can control the smart router to download 1080 P videos at high speed or play a TV show in real time. You can also control the smart switch to adjust low-data sensor-based devices for entertainment, movie, and sleep modes [3,4,5]. Big data scenarios challenge the current network architectures and network ability.

For indoor low-speed data services, the IoTs can interconnect and control intelligent applications through single-point contacts, such as positioning services, warehousing and transportation monitoring, and intelligent-based home furnishings. Although the wireless medium transmission has low costs and easy deployment advantages, there are problems, such as electromagnetic interference and resource shortages. Visible light communication (VLC) by ceiling LEDs can overcome the above issues and is an effective solution [6,7].

For indoor high-speed data services, traffic with high peak data rates frequently features latency and reliability requirements, such as the voice and video services in the eMBB scenario [8]. The reliability in regard to latency is a statistical quality of service (QoS) index, which indicates the probability of system latency within a threshold. Under the strict reliability requirements constraint, the traffic possessing bursty random characteristics and high data rates desires an amazingly large bandwidth, which challenges the insufficient underlying resources [9]. VLC is expected to be one of the supplements to relieve the tension of radio frequency (RF) resources.

We show the visible-light-based system applications for communication and positioning, as shown in Figure 1. The network data in the access network are broadcast at high speed through the LEDs, and the laptops or other devices receive the data through the high-response photodiode. The LEDs transmit a low-data sensor acquisition index or location coordinate landmark to achieve effective interaction among smart devices.

However, the visible-light-based application system still has the following problems:(1)For indoor low-speed data services, the system finds it difficult to extract distinguishable positioning landmark features without affecting communication performance when the visible-light-based system tries to achieve positioning and communication simultaneously, and the lost data bits in the interframe gap (IFG) have an effect on the system performance [10,11].(2)For indoor high-speed data services, a precise analysis framework of reliability in regard to latency is the crux for reasonable resource allocation, which should reveal the relationships among traffic characteristics, network abilities, and reliability requirements. The existing theoretical methods, such as stochastic network calculus and the effective bandwidth, appear to have mathematical barriers. For traffic with burstiness, these frameworks only provide loose upper bounds of unreliability in regard to latency [12,13]. It could trigger resource waste when theoretical results guide the reliability provisioning.

Here, in this paper, we design and implement a novel visible-light-based fusion application system, named the Fasys, to solve the above problems and achieve effective communication and positioning services. Specifically, when the LED is used to transmit low-speed communication data and positioning landmarks we find a robust linear mapping relationship between the data transmission rate and the width of the bright and dark stripes in the LED contour sub-maps. We only need to capture one complete sequence header instead of two sequence headers to recognize a specific LED. We also propose a linear block coding and bit interleaving mechanism. We encode the transmitted communication data according to the Hamming $\left(7,4\right)$ rule and obtain codewords named Hamming blocks. We perform the above mechanisms on the fixed number of Hamming blocks, which decreases the system’s image processing overhead and decreases the interframe gap. For service traffic with high data rates and burstiness, VLC is considered to provide great assistance for the RF link to guarantee reliability. A precise analysis method of reliability in regard to latency is proposed based on martingale theory. A backlog martingale process for the bursty traffic is constructed and the complex parameters are captured. Leveraging stopping time theory, a tight upper bound of unreliability is derived. Based on the theoretical result, the abstracted arrivals are determined for the RF link and the VLC link, i.e., the maximum arrival rates each link can support. An effective traffic allocation scheme is designed to achieve the provisioning of reliability in regard to latency.

In summary, our contributions are summarized as follows:We first design and implement a visible-light-based fusion applications system, Fasys, which can achieve concurrent communication and positioning fusion services.For the low-speed data services, we propose the novel linear block coding and bit interleaving mechanism, which enhances the LED positioning accuracy and recovers the lost data bits in the IFG.For the high-speed data services, an elegant reliability evaluation framework is proposed based on martingale construction for the traffic with burstiness. An upper bound of unreliability in regard to latency is obtained to guide the design of a traffic allocation scheme, which facilitates to decouple the statistical reliability requirement as the maximum arrival load the link can carry.We evaluate the effectiveness of the Fasys system via extensive experiments, and the results show that our Fasys system can achieve centimeter-level positioning accuracy and a Kbps communication rate for the low-speed data services. Meanwhile, the martingale-based traffic allocation scheme can provision the reliability requirement for the bursty traffic. The theoretical result we derive of the unreliability in regard to latency is precise.

## 2. The Communication and Positioning for Low-Speed Data Applications

In this section, we describe the low-speed data services, including the visible light positioning and the low-speed data visible light communication.

### 2.1. Visible-Light-Based Positioning

For the visible-light-based positioning part, it is important to design the LED’s mapping mechanism between the transmitter and the receiver to recognize the LED landmark accurately.

Compared with previous visible light positioning solutions, we do not use modulation and driving circuits to control each LED to transmit a unique identification code. Instead, we seek to extract distinguishable positioning landmark features from visible light signals (communication data) to set the mapping mechanism between the transmitter and the receiver.

Through extensive experiments, we found that when the transmitted data sequence period changes at a fixed frequency in the transmitter, the light and dark stripe widths are stable changes at captured LED sub-maps in the receiver. We use this robust linear mapping mechanism to transmit the positioning landmark and low-speed communication data. First, we ensure that the data communication frequencies $\left\{f_{1},f_{2},\ldots ,f_{n},\ldots \right\}$ are between 200 Hz and 8 KHz to avoid human eyes perceiving flickers. For the different LEDs $\left\{LED_{1},LED_{2},\ldots ,LED_{n},\ldots \right\}$, we assign a unique transmission frequency for each LED, so that the receiver can capture different sequence header widths $\left\{W_{1},W_{2},\ldots ,W_{n},\ldots \right\}$ in the received LED sub-maps. Then, for the assigned transmission frequencies, we ensure that each received sequence header width $W_{i},i=1,\;2,\;3,\ldots$ between captured LED sub-maps does not overlap. The detailed frequency design for the mapping mechanism is shown in Table 1. Finally, we use the different sequence header widths to match LED transmission frequencies and recognize specific LED landmarks rather than using the decoding LED sequence method.

Through the above mapping mechanism, we only need to match the width values of the captured sequence headers with the benchmark width values in the database to recognize the specific LEDs. It means that we can use the frequency of the transmission communication data to represent the LED landmarks, which does not have an effect on serving the communication data content.

### 2.2. Low-Speed Visible-Light-Based Communication

For the visible-light-based communication part, it is important to design a robust coding and error correction mechanism to solve the data loss problem in the interframe gap at the receiver.

Compared with previous visible light communication solutions, we do not consecutively transmit each communication data packet twice, which achieves a low BER by decreasing the communication data rate. Instead, we seek to transmit each data packet only once and design reasonable coding and error correction mechanisms to recover the lost data in the IFG. The communication data packet formats from our predecessor and our own proposal are shown in Figure 2a,b respectively.

We use the linear block coding and bit interleaving scheme to encode communication packets. First, we convert the communication data into ASCII codes and use the (7,4) Hamming coding rule to encode the converted ASCII codes, as shown in Figure 3a. Since linear block coding can only correct the one-bit errors, we then perform the bit interleaving on the transformed linear block coding, in order to spread possible errors to different Hamming blocks. Finally, we use the Manchester encoding rule to encode the transformed bit interleaving codes again, for example, the “1010” bits are converted the “10011001” bits, as shown in Figure 3b. For the decoding process at the receiver, we first insert some random codes between any two consecutive frames. Then, we perform Manchester decoding and bit de-interleaving on the current data, which disperses possible bit errors into different Hamming blocks. Finally, we use the (7,4) Hamming code to restore the received communication data.

In addition, the number of Hamming blocks determines the error correction performance of the communication transmission. We need to configure a reasonable number of Hamming blocks in the encoded data. The number of Hamming blocks is affected by (1) the amount of data lost in the interframe gap and (2) the packet capacity. According to the rolling shutter effect, the number of scanning lines $L_{b}$ used to receive data bits on the image sensor is as follows:(1)$$L_{b}=\left(L_{f}\times f\right)/R_{b}$$
where $L_{f}$ is the number of scan lines per frame, $R_{b}$ is the data rate, and *f* is the frame rate. The data capacity $N_{bpf}$ per frame can be calculated as follows:(2)$$N_{bpf}=L_{f}/L_{b}=R_{b}/f$$

Please note that, in order to achieve consistent visible light positioning and visible light communication, we need to ensure that the data packet capacity is as close as possible to the data capacity per frame. The specific constraint relationship is $N_{bpb}<N_{bpf}$. $N_{bpb}$ is the communication data packet capacity.

Through the linear block coding and bit interleaving mechanism, we only need to transmit the communication data packets once to realize the consistent transmission for low-speed visible light communication and visible light positioning.

## 3. The Reliability Provisioning for High-Speed Data Applications

In this section, we describe the high-speed data services. We assume that these kinds of applications possess statistical latency and reliability requirements. The reliability in regard to latency is called latency-bounded reliability in this paper. For the data traffic with latency-bounded reliability requirements, we design a traffic allocation scheme in a VLC/radio frequency (RF) heterogeneous network (HetNet). The VLC network is used to aid the traffic offloading for the RF network.

### 3.1. The Network Model and the Queuing System

An indoor downlink HetNet scenario is shown in Figure 4. The access network for the target user equipment (UE) consists of one RF access point (AP) and one VLC AP. The data traffic delivered to the target UE is split into two flows transmitted on the RF link and the VLC link, respectively. The edge software defined network (SDN) controllers could provide traffic allocation decisions.

The orthogonal frequency division multiple access (OFDMA) scheme is adopted in the RF network. The bandwidth of each RB is $W_{RF}$. hr(t) denotes the channel gain and pr(t) is the transmission power assigned to the UE on a resource block (RB) at time slot *t*. We use α to denote the RB allocation factor. α is the number of RBs that are allocated to the target UE and is a random variable obeying a specifical independent identical distribution (i.i.d.). Therefore, the achievable transmission rate of UE at time slot *t* is
(3)$$R_{RF}\left(t\right)=\alpha W_{RF}\log_{2}\left(1+\frac{pr\times hr(t)}{{\sigma_{RF}}^{2}}\right)\;\left[bits/s\right]$$
where ${\sigma_{RF}}^{2}$ is the noise power.

In the HetNet, we consider the light of sight (LoS) VLC link between the VLC AP and the UE. Then, the channel gain can be modeled as
(4)$$\begin{array}{c} hv=\frac{(b+1)Ap}{2\pi d^{2}}\cos^{b}\left(\varphi^{ir}\right)g_{of}f\left(\varphi^{in}\right)\cos\left(\varphi^{in}\right) \end{array}$$
where *b* is the Lambertian index, and $b=-1/\log_{2}\left(\cos\left(\vartheta_{1/2}\right)\right)$. $\vartheta_{1/2}$ is the half-intensity radiation angle. Ap is the photodiode (PD) area. *d* is the distance between AP *i* and UE *m*. $g_{of}$ is the gain of optical filter. $\varphi^{in}$ and $\varphi^{ir}$ are the angle of incidence and irradiance between the VLC AP and the UE, respectively. $f(\varphi^{in})$ is the optical concentrator gain, which is a function of $\varphi^{in}$
(5)$$f\left(\varphi^{in}\right)=\left\{\begin{array}{c} \frac{r^{2}}{\sin^{2}\left(\Theta \right)},\;0\leq \varphi^{in}\leq \Theta \\ 0,\;\varphi^{in}>\Theta \end{array}\right.$$
where *r* is the refractive index, and Θ is the semi-angle of the field of view (FoV) of the PD. According to [14], the electronic transmission power $pv^{e}\left(t\right)$ and optical transmission power $pv^{o}\left(t\right)$ when an ACO-OFDM scheme is used satisfy $pv^{e}\left(t\right)=\pi {\left(pv^{o}\left(t\right)\right)}^{2}$, where π is the responsivity of the PD. The achievable transmission rate for the UE, which is defined as $R_{VLC}\left(t\right)$, can be given as
(6)$$\begin{array}{c} R_{VLC}\left(t\right)=W_{VLC}\log_{2}\left(1+\frac{\rho^{2}pv^{e}\times hv^{2}}{{\sigma_{VLC}}^{2}}\right)\left[bits/s\right] \end{array}$$
where $W_{VLC}$ is the system bandwidth, and ρ is the optical to electrical (O/E) conversion factor. ${\sigma_{VLC}}^{2}$ is the noise power.

The RF link and VLC link are modeled as two queuing systems. Since latency is often defined from the perspective of packets, the arrival rate and service rates are sampled in the units of packets/slots instead of bits/s. $\left\{a_{RF}\left(t\right),t\geq 0\right\}$ and $\left\{a_{VLC}\left(t\right),t\geq 0\right\}$ denote the arrival processes of the two links, respectively. $a_{l}\left(t\right),l\in \left\{RF,VLC\right\}$ is the number of arrival packets at slot *t*. $s_{l}\left(t\right),\;l\in \left\{RF,VLC\right\}$ is the number of served packets in corresponding links at slot *t*. $s_{l}\left(t\right)=\left\lfloor \frac{fR_{l}\left(t\right)}{g}\right\rfloor$, where *f* (s/slot) is the duration of a slot and *g* (bits/packet) is the length of a packet. $R_{l}\left(t\right)\in \left\{R_{RF}\left(t\right),R_{VLC}\left(t\right)\right\}$ is the achievable transmission rate of the RF link or the VLC link. To maintain the stability of the queuing system, $E\left[a_{l}\left(t\right)\right]\leq E\left[s_{l}\left(t\right)\right]$. The accumulated arrival process $\left\{A_{l}\left(m,n\right),0\leq m\leq n\right\}$ and the accumulated service process $\left\{S_{l}\left(m,n\right),0\leq m\leq n\right\}$ from time slot *m* to *n* are driven as $A_{l}\left(m,n\right)=\sum_{k=m}^{n}a_{l}\left(k\right)$ and $S_{l}\left(m,n\right)=\sum_{k=m}^{n}s_{l}\left(k\right)$, respectively. The total number of data packets delivered to the target UE is $A\left(m,n\right)=A_{RF}\left(m,n\right)+A_{VLC}\left(m,n\right)$. Based on stochastic network calculus (SNC) theory, the departure process $D_{l}\left(0,n\right)$ can be characterized by $A_{l}\left(0,n\right)$ and $S_{l}\left(0,n\right)$.
(7)$$\begin{array}{cc} D_{l}\left(0,\;n\right) & =A_{l}⊗S_{l}\left(0,\;n\right)=\underset{0\leq k\leq n}{\inf}\left\{A_{l}\left(0,\;k\right)+S_{l}\left(k,\;n\right)\right\} \end{array}$$
where the processes of arrival, service, and departure are connected by the (min, +) convolution. Based on the queuing theory, the backlog process $Q_{l}\left(t\right)$ of link l,l∈{RF,VLC} in the buffer is defined as
(8)$$Q_{l}\left(t\right)=\underset{0\leq m\leq t}{\sup}\left\{A_{l}\left(m,\;t\right)-S_{l}\left(m,\;t\right)\right\}$$

And the latency of link l,l∈{RF,VLC} at time slot *t* is given as
(9)$$W_{l}\left(t\right)=\min\left\{k\geq 0|A_{l}\left(0,\;t-k\right)\leq D_{l}\left(0,\;t\right)\right\}$$

The statistical latency-bounded reliability requirement includes two metrics: the latency threshold $W^{\max}$ and the bound of unreliability ζ. The traffic allocation pattern is adopted in this paper. The arrival flows allocated to the RF link and VLC link should both be provisioned reliably. That is, $W_{l}\left(t\right),l\in \left\{RF,VLC\right\}$ should converge to its stationary distribution, which means that the latency is required to ensure $P\{W_{l}\left(t\right)\geq W^{\max}\}\leq \zeta .$

The traffic load that each link can carry depends on the corresponding channel state and the access scheme. The determination of the ideal traffic load needs the precise analysis of the link latency-bounded reliability. We define link latency-bounded reliability as the probability that the link latency does not exceed the predefined latency threshold $W_{l}^{\max}$, a lower bound of which is denoted as $RE_{l}\left(W_{l}^{\max},a_{l}\left(t\right),s_{l}\left(t\right)\right),l\in \left\{RF,VLC\right\}$. It is a function of the latency threshold, the characteristics of the arrival process $\left\{a_{l}\left(t\right),t\geq 0\right\}$, and the characteristics of the service process $\left\{s_{l}\left(t\right),t\geq 0\right\}$. We analyze $RE_{l}\left(W_{l}^{\max},a_{l}\left(t\right),s_{l}\left(t\right)\right)$ leveraging martingale theory, which contributes to describe the randomness of the queue length (the difference of the arrival process and the service process). In the martingale domain, the inequalities in regard to martingales support the unreliability analysis naturally. We can obtain an elegant closed form of $RE_{l}\left(W_{l}^{\max},a_{l}\left(t\right),s_{l}\left(t\right)\right),l\in \left\{RF,VLC\right\}$, finally.

The analysis procedure of the link latency-bounded reliability can be concluded as follows: (1) The backlog process $Q_{l}\left(t\right)$ is constructed as an exponent martingale, which is denoted as $M_{l}^{L}\left(t\right)=f_{l}\left(t;a_{l}\left(t\right),s_{l}\left(t\right)\right)$. $M_{l}^{L}\left(t\right)$ is the function of $Q_{l}\left(t\right)$; (2) We focus on a special event SW about link latency, i.e., $SW=\{W_{l}\left(t\right)\geq W_{l}^{\max}\}$; (3) By applying stopping time theory to $M_{l}^{L}\left(t\right)$, the probability of stopping time event occurrence, P{SW}, can be derived. That is, $1-RE_{l}\left(W_{l}^{\max},a_{l}\left(t\right),s_{l}\left(t\right)\right)$ is obtained ($P\left\{SW\right\}\leq 1-RE_{l}\left(W_{l}^{\max},a_{l}\left(t\right),s_{l}\left(t\right)\right)$).

In the traffic allocation phase, for the traffic flow allocated to link l∈{RF,VLC}, the reliability requirement should be met, i.e., $RE_{l}\left(W_{l}^{\max},a_{l}\left(t\right),s_{l}\left(t\right)\right)=1-\zeta$. Based on the theoretical result we derived, the ideal traffic load of each link can be determined.

Next, the novel martingale analysis method of the link latency performance, the arrival abstraction, and the traffic allocation scheme are introduced.

### 3.2. The Martingale-Based Link Latency-Bounded Reliability

Some definitions about martingales are presented first.

**Definition** **1.**
*(Martingales): The discrete time stochastic sequence {X(t),t≥0} is a martingale with respect to another sequence {Y(t),t≥0} if the following applies for ∀t≥0:   *
*(a)* 

E[|X(t)|]<∞;

*(b)* 
*E[X(t+1)|Y(0),Y(1),⋯,Y(t)]=X(t).*



We model the generation process of data packets in the target UE as a Markov-modulated random process to describe the bursty characteristics. Thus, the traffic flows allocated to the two links are also depicted as the same processes, merely with different arrival parameters. In the following, a backlog martingale for the backlog process of each link is constructed.

**Definition** **2.**
*(Backlog martingale). Let $\{Z_{l}\left(t\right),t\geq 0\},l\in \left\{RF,VLC\right\}$ be a Markov process with the state space $T_{l}=\left\{0,\;1\right\}$ and the transition matrix is $\Phi_{l}$. $f_{l}:S\to R^{+}$ is a deterministic function. The arrival $a_{l}\left(t\right)=f_{l}\left(Z_{l}\left(t\right)\right)=R_{l}^{max}Z_{l}\left(t\right)$. $R_{l}^{max}$ is the maximum peak arrival rate that link l can support. The service process is $\left\{s_{l}\left(t\right),t\geq 0\right\}$. For $\theta_{l}^{*}$, $j_{l}$, and $\gamma_{l}$, there is a process $\left\{M_{l}^{L}\left(t\right),\;t\geq 0\right\}$ about $\{Q_{l}\left(t\right),\;t\geq 0\},$*

(10)
$$M_{l}^{L}\left(t\right)=V_{l}^{a}\left(z_{l}\left(t\right)\right)V_{l}^{s}\left(s_{l}\left(t\right)\right)e^{\theta^{*}\left[A_{l}\left(m,t\right)-S_{l}\left(m,t\right)\right]}$$

*which is constructed as a martingale process. Then, $\left\{M_{l}^{L}\left(t\right),\;t\geq 0\right\}$ is called a backlog martingale. The parameter $\theta^{*}$ is defined as*

(11)
$$\theta^{*}=\sup\left\{\theta >0:sp\left(\Phi_{l}^{\theta }\right)E\left[e^{-\theta s_{l}\left(t\right)}\right]=1\right\}$$

*where sp(x) is the spectral radius of x. $\Phi_{l}^{\theta }$ is defined as a θ−transform for $\Phi_{l}$. If ${\left(x\right)}_{i,j}$ is the element of x, then ${\left(\Phi_{l}^{\theta }\right)}_{i,j}={\left(\Phi_{l}\right)}_{i,j}e^{\theta f_{l}\left(Z_{l}\left(t+1\right)\right)}$. $V_{l}^{a}$ is the corresponding eigenvector of $sp(\Phi_{l}^{\theta })$. $V_{l}^{a}\left(z_{l}\left(t\right)\right)$ is the $\left(z_{l}\left(t\right)\right)$th element of $V_{l}^{a}$. $V_{l}^{s}\left(s_{l}\left(t\right)\right)$ is a specifical function of service process $\left\{s_{l}\left(t\right),t\geq 0\right\}$.*


The proof of why Equation (Equation 10) is a martingale process relies on the definition of martingales.

**Proof.** Due to the stability assumption, $E\left[M_{l}^{L}\left(t\right)\right]=E[|M_{l}^{L}\left(t\right)\left|\right]<\infty$ obviously holds. And then Equation (Equation 12) can be derived. (12a) holds because the arrival and service processes are independent, and $\{a_{l}\left(t\right),t\geq 0\},l\in \left\{RF,VLC\right\}$ is the Markov-modulated random process. (12b) relies on the definition of conditional expectation and features of i.i.d. service processes. (12c) is from the definition of $\Phi_{l}^{\theta }$. The relation between $sp(\Phi_{l}^{\theta })$ and $V_{l}^{a}$ supports (12d) to (12e). Leveraging the defined $\theta^{*}$, the last step (12f) is finished. Thus, $E\left[M_{l}^{L}\left(t+1\right)|Q_{l}(0),Q_{l}(1),\cdots ,Q_{l}\left(t\right)\right]=M_{l}^{L}\left(t\right)$ holds. The proof is completed.
(12)$$\begin{array}{c} E[M_{l}^{L}\left(t+1\right)|Q_{l}(0),Q_{l}(1),\cdots ,Q_{l}\left(t\right)]\\ =E[V_{l}^{a}\left(z_{l}\left(t+1\right)\right)V_{l}^{s}\left(s_{l}\left(t+1\right)\right)e^{\theta^{*}\left[A_{l}\left(m,t+1\right)-S_{l}\left(m,t+1\right)\right]}|Q_{l}(t)]\\ =E\left[V_{l}^{a}\left(z_{l}\left(t+1\right)\right)e^{\theta^{*}A_{l}\left(m,t+1\right)}|z_{l}(t)\right]E\left[V_{l}^{s}\left(s_{l}\left(t+1\right)\right)e^{-\theta^{*}S_{l}\left(m,t+1\right)}|s_{l}(t)\right]\begin{array}{cc} & \left(12a\right) \end{array}\\ =e^{\theta^{*}A_{l}\left(m,t\right)}\sum_{z_{l}\left(t+1\right)\in T_{l}}V_{l}^{a}\left(z_{l}\left(t+1\right)\right)e^{\theta^{*}f_{l}\left(z_{l}\left(t+1\right)\right)}P\left\{z_{l}\left(t+1\right)\right|z_{l}(t)\}V_{l}^{s}\left(s_{l}\left(t\right)\right)e^{-\theta^{*}S_{l}\left(m,t\right)}E\left[e^{-\theta^{*}s_{l}\left(t\right)}\right]\begin{array}{cc} & \left(12b\right) \end{array}\\ \overset{z_{l}\left(t\right)=i,z_{l}\left(t+1\right)=j}{=}V_{l}^{s}\left(s_{l}\left(t\right)\right)e^{\theta^{*}\left[A_{l}\left(m,t\right)-s_{l}\left(t+1\right)\right]}E\left[e^{-\theta^{*}s_{l}\left(t\right)}\right]\sum_{j\in T_{l}}V_{l}^{a}\left(j\right){\left(\Phi_{l}^{\theta }\right)}_{i,j}\begin{array}{cc} & \left(12c\right) \end{array}\\ =V_{l}^{s}\left(s_{l}\left(t\right)\right)e^{\theta^{*}\left[A_{l}\left(m,t\right)-s_{l}\left(t+1\right)\right]}E\left[e^{-\theta^{*}s_{l}\left(t\right)}\right]\left(\Phi_{l}^{\theta }V_{l}^{a}\right)\left(i\right)\begin{array}{cc} & \left(12d\right) \end{array}\\ =V_{l}^{s}\left(s_{l}\left(t\right)\right)e^{\theta^{*}\left[A_{l}\left(m,t\right)-s_{l}\left(t+1\right)\right]}E\left[e^{-\theta^{*}s_{l}\left(t\right)}\right]sp\left(\Phi_{l}^{\theta }\right)V_{l}^{a}\left(z_{l}\left(t\right)\right)\begin{array}{cc} & \left(12e\right) \end{array}\\ =M_{l}^{L}\left(t\right)\begin{array}{cc} & \left(12f\right) \end{array} \end{array}$$$R_{l}^{max}$ is called the abstracted arrival. The abstracted arrival is a measure of the traffic load that each link can carry, considering the stochastic characteristics of the traffic flow and the service scheme of link *l*. The determination of $R_{l}^{max}$ relies on evaluating the link latency performance precisely. Based on the backlog martingale, we can yield the bound of backlog overflow probability via stopping time theory in the martingale domain. The backlog overflow probability is given by Theorem 1.    □

**Theorem** **1.**
*Consider a queuing system corresponding to the backlog martingale as Definition 2. The following bound of backlog overflow probability holds for any backlog threshold $\eta_{l}\geq 0$*

(13)
$$P\left\{Q_{l}\left(t\right)\geq \eta_{l}\right\}\leq \frac{E\left[V_{l}^{a}\left(z_{l}\left(0\right)\right)\right]E\left[V_{l}^{s}\left(s_{l}\left(0\right)\right)\right]}{H}e^{-\theta^{*}\eta_{l}}$$

*where*

(14)
$$H=\min\{V_{l}^{a}\left(z_{l}\left(t\right)\right)V_{l}^{s}\left(s_{l}\left(t\right)\right):a_{l}\left(t\right)-s_{l}\left(t\right)>0\}$$



**Proof.** The proof of Theorem 1 is based on stopping time theory. A stopping time event about $\left\{Q_{l}\left(t\right),\;t\geq 0\right\}$ is defined as $SL=\{Q_{l}\left(t\right)\geq \eta_{l}\}$. Then, the first time that SL occurs is the stopping time of the backlog martingale $\left\{M_{l}^{L}\left(t\right),\;t\geq 0\right\}$, which is written as
(15)$$N_{L}=\inf\left\{t>0:A_{l}\left(0,t\right)-S_{l}\left(0,t\right)\geq \eta_{l}\right\}$$According to the stopping time theory of martingales, we have $E\left[M_{l}^{L}\left(0\right)\right]=E\left[M_{l}^{L}\left(t\wedge N_{L}\right)\right]$, where $t\wedge N_{L}=\min\left\{t,N_{L}\right\}$. Thus, it can be derived that
(16)$$\begin{array}{cc} \; & E\left[M_{l}^{L}\left(0\right)\right]=E\left[V_{l}^{a}\left(z_{l}\left(0\right)\right)V_{l}^{s}\left(s_{l}\left(0\right)\right)\right]\\ \; & =E[V_{l}^{a}\left(z_{l}\left(t\wedge N_{L}\right)\right)V_{l}^{s}\left(s_{l}\left(t\wedge N_{L}\right)\right)\times e^{\theta^{*}\left[A_{l}\left(0,t\wedge N_{L}\right)-S_{l}\left(0,t\wedge N_{L}\right)\right]}]\\ \; & \geq E[V_{l}^{a}\left(z_{l}\left(N_{L}\right)\right)V_{l}^{s}\left(s_{l}\left(N_{L}\right)\right)\times e^{\theta^{*}\left[A_{l}\left(0,N_{L}\right)-S_{l}\left(0,N_{L}\right)\right]}1_{\left\{N_{L}\leq t\right\}}]\\ \; & \geq He^{\theta^{*}\eta_{l}}P\left\{N_{L}\leq t\right\} \end{array}$$By minimizing $N_{L}$, $a_{l}\left(N_{L}\right)-s_{l}\left(N_{L}\right)>0$ and $E[V_{l}^{a}\left(z_{l}\left(N_{L}\right)\right)V_{l}^{s}\left(s_{l}\left(N_{L}\right)\right)>H$ hold. It is possible that $P\left\{N_{L}\leq \infty \right\}=P\left\{Q_{l}\left(t\right)\geq \eta_{l}\right\}$. The proof is completed when $N_{L}\to \infty$.Based on the bound of backlog overflow probability, the bound of link latency-bounded unreliability is derived in Theorem 2.    □

**Theorem** **2.**
*In the case of Theorem 1, the following bound of latency $W_{l}\left(t\right)$ holds for*

(17)
$$\begin{array}{cc} \; & P\left\{W_{l}\left(t\right)\geq W_{l}^{\max}\right\}\leq \frac{E\left[V_{l}^{a}\left(z_{l}\left(0\right)\right)\right]E\left[V_{l}^{s}\left(s_{l}\left(0\right)\right)\right]}{H}e^{-\theta^{*}W_{l}^{\max}\frac{\ln sp(\Phi_{l}^{\theta^{*}})}{\theta^{*}}} \end{array}$$



**Proof.** (18)$$\begin{array}{cc} \; & P\{W_{l}\left(t\right)\geq W_{l}^{\max}\}\\ \; & =P\{A_{l}\left(0,t-W_{l}^{\max}\right)\geq D_{l}\left(0,t\right)\}\\ \; & \leq P\{A_{l}\left(0,t-W_{l}^{\max}\right)\geq \underset{0\leq m\leq t}{\inf}\left\{A_{l}\left(0,m\right)+S_{l}\left(m,t\right)\right\}\}\\ \; & \leq P\{\underset{0\leq m\leq t-W_{l}^{\max}\leq t}{\sup}\left\{A_{l}\left(m,t-W_{l}^{\max}\right)-S_{l}\left(m,t\right)\right\}\geq 0\}\\ \; & =P\{\underset{0\leq m\leq t-W_{l}^{\max}\leq t}{\sup}\{A_{l}\left(m,t-W_{l}^{\max}\right)-S_{l}\left(m,t\right)\\ \; & \;\;+\left(t-W_{l}^{\max}-m\right)\frac{\ln E{\left[e^{-\theta^{*}s_{l}\left(t\right)}\right]}^{-1}-\ln sp\left(\Phi_{l}^{\theta^{*}}\right)}{\theta^{*}}\}\geq \frac{\ln sp(\Phi_{l}^{\theta^{*}})}{\theta^{*}}W_{l}^{\max}\} \end{array}$$
The last step relies on the definition of $\theta^{*}$. $\ln E{\left[e^{-\theta^{*}s_{l}\left(t\right)}\right]}^{-1}-\ln sp\left(\Phi_{l}^{\theta^{*}}\right)=0$ holds. Based on the time shift feature of martingales, it is obvious that
(19)$$\begin{array}{cc} M_{l}^{L-s}\left(t\right) & =V_{l}^{a}\left(z_{l}\left(t-W_{l}^{\max}\right)\right)V_{l}^{s}\left(s_{l}\left(t\right)\right)e^{\theta^{*}\left[A_{l}\left(m,t-W_{l}^{\max}\right)-S_{l}\left(m,t\right)\right]} \end{array}$$
is also a martingale process. To complete the latency analysis, the stopping time event $SW=\{W_{l}\left(t\right)\geq W_{l}^{\max}\}$ can be transformed into  
(20)$$\begin{array}{cc} SW & =\{A_{l}\left(m,t-W_{l}^{\max}\right)-S_{l}\left(m,t\right)\\ \; & +\left(t-W_{l}^{\max}-m\right)\frac{\ln E{\left[e^{-\theta^{*}s_{l}\left(t\right)}\right]}^{-1}-\ln sp\left(\Phi_{l}^{\theta^{*}}\right)}{\theta^{*}}\geq \frac{\ln sp(\Phi_{l}^{\theta^{*}})}{\theta^{*}}W_{l}^{\max}\} \end{array}$$Then, the first time slot that SW occurs in is defined as the stopping time $N_{W}$.
(21)$$\begin{array}{cc} N_{W} & =\inf\{t>0:A_{l}\left(m,t-W_{l}^{\max}\right)-S_{l}\left(m,t\right)\\ \; & +\left(t-W_{l}^{\max}-m\right)\frac{\ln E{\left[e^{-\theta^{*}s_{l}\left(t\right)}\right]}^{-1}-\ln sp\left(\Phi_{l}^{\theta^{*}}\right)}{\theta^{*}}\geq \frac{\ln sp(\Phi_{l}^{\theta^{*}})}{\theta^{*}}W_{l}^{\max}\} \end{array}$$Renting the stopping time theory for the martingale process $M_{l}^{L-s}\left(t\right)$, we can derive that
(22)$$\begin{array}{cc} \; & E\left[M_{l}^{L-s}\left(0\right)\right]=E\left[V_{l}^{a}\left(z_{l}\left(0\right)\right)V_{l}^{s}\left(s_{l}\left(0\right)\right)\right]\geq He^{\theta^{*}\frac{\ln sp(\Phi_{l}^{\theta^{*}})}{\theta^{*}}W_{l}^{\max}}P\left\{N_{W}\leq t\right\} \end{array}$$The parameter *H* is defined as (Equation 14). When $N_{W}\to \infty$, we can obtain $P\left\{W_{l}\left(t\right)\geq W_{l}^{\max}\right\}=P\left\{N_{W}\geq \infty \right\}$ as Theorem 2. Thus, we obtain the latency-bounded reliability of link l,l∈{RF,VLC}, which is derived as
(23)$$\begin{array}{cc} \; & RE_{l}\left(W_{l}^{\max},a_{l}\left(t\right),s_{l}\left(t\right)\right)=1-\frac{E\left[V_{l}^{a}\left(z_{l}\left(0\right)\right)\right]E\left[V_{l}^{s}\left(s_{l}\left(0\right)\right)\right]}{H}e^{-\theta^{*}W_{l}^{\max}\frac{\ln sp(\Phi_{l}^{\theta^{*}})}{\theta^{*}}} \end{array}$$   □

### 3.3. The Arrival Abstraction and Traffic Allocation Scheme

Based on the derived bound of link unreliability in regard to latency, we designed Algorithm 1 to achieve the arrival abstraction and the traffic allocation. The process is summarized in Figure 5. The input of the algorithm includes the arrival model, the service model of each link, and the corresponding model parameters. For the total arrival traffic {a(t),t≥0}, the peak arrival rate is denoted as $R^{max}$. The backlog process and the latency process are described by accumulated arrival and accumulated service processes. In the martingale domain, the arrival-related martingale parameter $\ln sp(\Phi_{l}^{\theta })$ and the service-related martingale parameter $E[e^{-\theta s_{l}\left(t\right)}]$ should be determined. Accordingly, the martingale parameter $\theta^{*}$ is obtained. We construct the backlog martingale for the backlog process. Relying on the stopping time theory, the upper bound of the unreliability in regard to link latency is derived. Using this theoretical analysis result, the abstracted arrival of each link, $R_{l}^{\max}$ is captured under the constraint of the latency-bounded reliability requirement. Finally, the traffic allocation decision is made according to the ratio of $R_{RF}^{\max}$ and $R_{VLC}^{\max}$. Unfortunately, the closed-form solution of abstracted arrival is unable to be obtained. Therefore, search algorithms, such as the binary search algorithm, can be adapted to obtain $R_{l}^{\max}$, as is shown in Algorithm 1.
**Algorithm 1** Arrival Abstraction for the RF and VLC links1:**Input:** The transition matrix $\Phi_{l}$. The latency threshold $W^{\max}$. The unreliability requirement ζ. The maximum tolerance $tol_{AA}$. The model parameters of the RF link, α, $W_{RF}$, hr(t), pr, ${\sigma_{RF}}^{2}$. The model parameters of the VLC link, ρ, $W_{VLC}$, hv, $pv^{e}$, ${\sigma_{VLC}}^{2}$.2:According to the characteristics of arrival and service processes, determine the martingale parameters $V_{l}^{a}\left(z_{l}\left(t\right)\right)$, $V_{l}^{s}\left(s_{l}\left(t\right)\right)$, $\theta^{*}$, respectively. Construct the backlog martingale process $\left\{M_{l}^{L}\left(t\right),\;t\geq 0\right\}$.3:Derive the link latency-bounded unreliability as shown in (Equation 17).4:**while** $|\zeta -\frac{E\left[V_{l}^{a}\left(z_{l}\left(0\right)\right)\right]E\left[V_{l}^{s}\left(s_{l}\left(0\right)\right)\right]}{H}e^{-\theta^{*}W_{l}^{\max}\frac{\ln sp(\Phi_{l}^{\theta^{*}})}{\theta^{*}}}|\leq tol_{AA}$**do**5:    Record the abstracted arrival $R_{RF}^{\max}$ and $R_{VLC}^{\max}$.6:**end while**7:**Output:** the abstracted arrival $R_{RF}^{\max}$ and $R_{VLC}^{\max}$.

## 4. Discussion

In this section, we evaluate our Fasys system through extensive experiments, for low-speed communication, positioning, and high-speed communication applications. In the low-speed communication and positioning application, we use the Cree XLamp CXA1512 as the LED for the transmitter, which can provide 900–1200 lm. Meanwhile, we use the STM32F407ZGT6 series microcontroller and IRF520 module as the system driving circuit. For the receiver, we use the google pixel smartphone to capture LED images, under 64,000–92,500 ns exposure times. The vertical distance between the LED and the smartphone is 0.8–1.1 m. In the high-speed communication application, we use Matlab to test the system performance, and all of the software parameters for high-speed services are shown in Table 2.

We use the image decoding rate, positioning success rate, and latency-bounded unreliability to test the system performance. The positioning success rate indicates the ratio of the number of captured images that can be successfully positioned to the total number of captured images. The image decoding rate indicates the ratio of the number of images decoded by all LEDs in the image to the total number of captured images. Please note that the captured image can contain 1–3 LEDs; we only need to recognize one LED to achieve positioning, but one LED does not mean that the image decoding rate in the captured image is 100%. For low-speed positioning and communication services, we use the ratio of the number of bits correctly decoded in captured images to the total number of bits transmitted by the transmitter to calculate the system BER. For high-speed communication services, the latency-bounded unreliability of the target UE is simulated to verify the effectiveness of the proposed arrival abstraction and traffic allocation scheme. The link latency-bounded unreliability is measured by the proportion of time slots in which the corresponding latency threshold is violated.

### 4.1. Visible-Light-Based Positioning and Low-Speed Communication

We test the positioning success rate and image decoding rate to test the visible-light-based positioning and low-speed communication stability and consistency. The test selects 5 × 5 positioning points from the positioning area centered on three LEDs. The positioning success rate and image decoding rate results are shown in Figure 6. From Figure 6, we can find that the data bit recognition rate in the image data for the positioning and communication is higher than 90%, and the positioning success rate is 100%. This shows that our system can stably provide low-rate visible light communication and high-recognition-rate landmark decoding.

We use the cumulative distribution function (CDF) to describe the positioning part in the low-speed service. The system receiver is placed in 100 random positions in the positioning area, and each position can capture at least one complete LED. The positioning accuracy results are shown in Figure 7. From Figure 7, we can find that, in the 100 sample data, 90% of the positioning error results are always less than 7.5 cm, which shows that our proposed positioning part can achieve centimeter-level positioning effects.

In addition, we use the BER to test the communication part in the low-speed service. Note that our communication transmission scheme only transmits each data packet once. We transmit 100,000 random bits under different SNR conditions, and the results are shown in Figure 8. From Figure 8 that we can find that our scheme has a lower data BER than the traditional communication scheme, which shows that the low-speed communication part can provide a good communication performance. Meanwhile, we only need to transmit each data packet once, which means that our transmission rate is higher than that of the traditional scheme.

In Figure 9, we show the impact of the number of the different Hamming blocks on the low-speed communication performance. From Figure 9 we can find that, when the number of Hamming blocks increases, the low-speed communication system has the lower BER. The number of the Hamming blocks shows that the bit interleaving mechanism can recover the lost data bits in the IFG effectively.

### 4.2. The Arrival Abstraction and Traffic Allocation Scheme for High-Speed Communication

We show the numerical results and analyze the performance of the proposed traffic allocation scheme.

The effectiveness of the proposed arrival abstraction and traffic allocation algorithm is illustrated firstly. We simulate an RF link and a VLC link. The RF AP and the VLC AP are located at the (0, 0, 2.5) coordinate point. The UE is at the (1, 1, 0.5) coordinate point. The path loss of the RF link is modeled as 31.5+35*log(Dr), where Dr is the distance between the RF AP and the UE. The RB allocation factor α follows a poisson distribution with average λ. Two queuing systems are constructed to describe the packets’ transmission processes, respectively. The arrival traffic is generated by the MMOO model. For a special MMOO arrival traffic with peak rate $R^{max}$, the arrival abstraction and traffic allocation algorithm is performed. And the corresponding data flows that two links should deliver are determined. We record the latency of each system. Because two independent links are used, for each slot, if the latency of any link exceeds the threshold, this slot is labeled as a violated slot. The averages of the unreliability are shown as a histogram in Figure 10. The unreliability requirement ζ can be guaranteed when the latency threshold is relaxed (5 slots to 10 slots). A stricter latency threshold means that more bandwidth resources should be provisioned so that the identical reliability requirement can be met. Based on the current link configuration, the guarantee of the reliability requirement for the MMOO arrival traffic is unachievable if the latency threshold is less than four slots.

However, using only a single link could make the reliability guarantee more terrible, as is shown in Figure 11. We simulate the link latency-bounded unreliability correspondingly on the case that the RF link or the VLC link delivers the traffic individually, where the green box-plots represent the results that the RF link only used and the blue box-plots otherwise. It shows that the reliability requirement can not be met in either case. Adopting only one link is incompetent. The system load is 0.4 (measured as the ratio of the average arrival rate to the average service rate), which implies that the system is light-load. However, the burstiness of the MMOO arrival traffic makes the two links unable to provision the reliability, even though the average service rate is up to 10 packets/slot. Meanwhile, the randomness of the service process in the RF link, which results from the RB allocation factor, causes the link latency-bounded unreliability to worsen. The service process of the VLC link can be modeled as a constant rate process. Thus, reducing the randomness of the service process as much as possible is the crux for the latency-bounded reliability provisioning.

The impact of the randomness in the service process on the abstracted arrival is analyzed and the results are shown in Figure 12. The abstracted arrival is the maximum peak rate that one link can support under the constraint of the latency-bounded reliability requirement if the arrival traffic is modeled as an MMOO process. When the service rates are constant, i.e., corresponding to the VLC link, the system could carry a greater arrival load than the random service process. Meanwhile, the reliability is also higher, as is shown in Figure 11. When the latency threshold is tight (one slot to four slots), the abstracted arrival is more sensitive to the randomness of service.

## 5. Related Work

In this section, we introduce the closest work for visible-light-based communication and positioning.

### 5.1. Visible-Light-Based Communication

In 2000, Tanaka et al. [15] proposed to use a white LED to build a wireless optical communication system for the first time, which creates a precedent for using an LED as a communication device. In 2003, the Visible Light Communication Consortium [16] was established in Japan, and the research on visible light communication gradually attracted the attention of scholars. In 2011, IEEE 802.15.7 [17] set the standards for short-distance wireless optical communication using visible light, including the specific design standardization for the data link layer and physical layer. The VLC has gained momentum as a great aid for the RF networks to provision the reliable delivery of various services. A link section strategy in hybrid VLC/RF networks was investigated in [18]. The non-asymptotic data backlog and delay violation probability bounds were obtained for the proposed three link usage strategies. The guarantee of statistical reliability in regard to latency is a bottleneck for the VLC network configuration and deployment, which need a precise reliability analysis framework as guidance.

Queuing theory is the basic method to analyze the latency performance, which can only provide the average results of latency [19,20]. Stochastic network calculus (SNC) is an important mathematical framework in terms of latency probabilistic bounds analysis. SNC contributes many fundamental concepts and operations, such as the definitions of departure, backlog, latency processes, and (min, +) convolutions. In [21], the upper bound of the unreliability in regard to latency was derived, where union bound inequality was adopted. It estimates an upper bound of the event probability, which facilitates the conversion of the (min, +) convolutions into the univariate convolution in classical algebra. Unfortunately, as the key enabler of latency performance analysis based on SNC, union bound inequality could induce a loose delay bound [22]. Effective bandwidth (EB) and effective capacity (EC) theories are adopted to analyze the latency performance widely. However, the assumption of constant fluid [12] and the usage of the union bound are drawbacks of this framework, which induce the derived bound of delay violation probability to be loose. For the bursty traffic, the bandwidth demands could be overestimated. As a powerful mathematical method, martingale theory is envisioned to present a more precise analysis result of latency performance. Researchers have testified that the bound of delay violation probability based on martingales improves the state-of-the-art bound by several orders of magnitude [23].

The research team in [22,23,24] was devoted to evaluating latency performance based on martingale theory. They proposed a modularity method to derive the upper bound of unreliability in regard to latency by a suitable exponential martingale transform of arrival and service processes. The authors in [25,26] expanded the martingale analysis framework in multi-hop vehicular ad hoc networks and the computation offloading scenario so that the system latency can be analyzed precisely. A differentiated ALOHA random access algorithm was proposed in [13], in which the energy efficiency maximization problem was formulated with the martingale-based delay violation probability bound constraints. In [27], a tight bound of delay violation probability for aggregate traffic was derived by leveraging supermartingale theory. These remarkable works guide our research.

### 5.2. Visible-Light-Based Positioning

Rahman et al. [28] proposed a binocular vision positioning system, which uses multi-LED image sensors at different imaging positions and the relative positions between different image sensors to obtain the information required for triangulation and then calculates the position of the receiver. Kim et al. [29] proposed the use of different colored LEDs to realize an indoor visible light positioning system. The system uses different colors to mark the world coordinate position, the receiver is two computer cameras, and the system calculates the receiver position through the images’ geometric relationship. Kuo et al. [30] proposed using solid-state lighting equipment and smartphone cameras to achieve accurate indoor positioning. They use the CMOS camera in the commercial smartphone to take LED images, and use the improved AOA algorithm for positioning, which solves the problem of using multiple image sensors. Hassan et al. [31] conducted research on the indoor positioning system based on visible light LEDs, and discussed and compared the existing visible light positioning systems. Meanwhile, they showed several new research, implementation, commercialization, and standardization directions for the future development of visible light positioning technology. Keskin et al. [32] used a distance estimation algorithm based on Time of Arrival (TOA) and Received Signal Strength (RSS) to estimate the effect of the sampling rate on the positioning error and proposed a distance estimation algorithm to improve the theoretical limit of positioning error for the synchronous and asynchronous visible light positioning systems. Rátosi et al. [33] proposed the dynamic visible light indoor positioning system, which can position and track moving objects. The under-sampling scheme provides high-frequency flicker-free operation for the landmark and uses cameras to capture image streams to estimate moving objects in real time. Yazar et al. [34] proposed a pulse optimization design method based on light-emitting diodes (LEDs) as transmitters. This method quantifies the positioning accuracy by the Cramér-Rao lower limit (CRLB) and studies the problem of maximizing the positioning performance of asynchronous and synchronous visible light positioning systems, which can reduce energy consumption by about 45% or improve positioning accuracy by 25%.

## 6. Conclusions

In this paper, for the big data applications with heterogeneous requirements, we have implemented a visible-light-based fusion application system, named Fasys, which can achieve high-speed and low-speed communication and high-precision positioning services, simultaneously. First, we proposed a linear block coding and bit interleaving mechanism for the low-speed data services, which enhances the LED positioning accuracy and recovers the lost data bits in the IFG. Then, we proposed an arrival abstraction and traffic allocation scheme for the high-speed data services in the VLC/RF HetNet. A precise analysis framework of reliability in regard to latency was constructed based on martingale theory. Finally, we evaluated our system performance, and the results show that the Fasys system can achieve consistent centimeter-level positioning and low-BER data communication. The obtained upper bound of the reliability in regard to latency is tight, and the traffic allocation algorithm can provide the reliability provisioning reasonably.

## Figures and Tables

**Figure 1 sensors-23-06340-f001:**
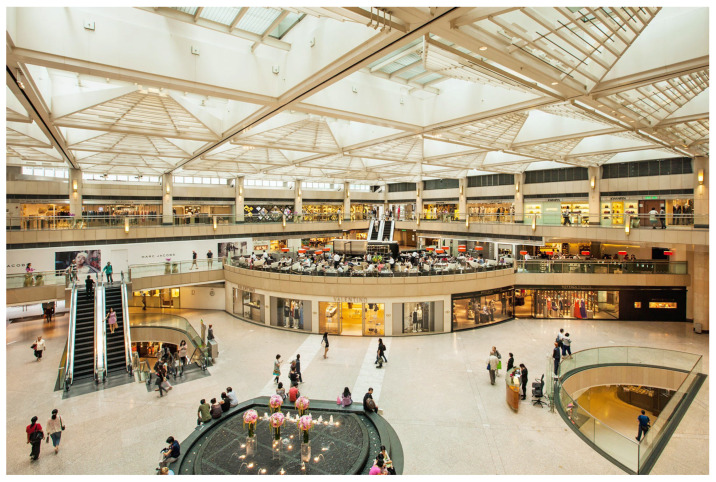
The visible-light-based communication and positioning for smart cities.

**Figure 2 sensors-23-06340-f002:**
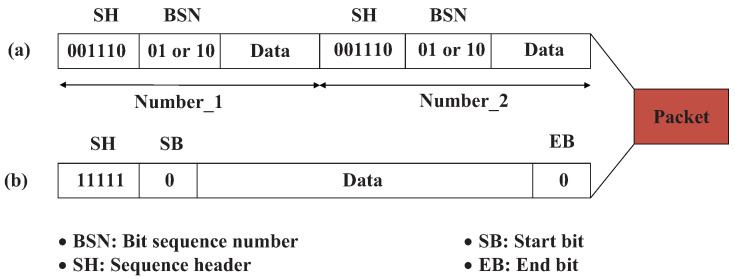
Data packet format comparison between the (**a**) predecessor and (**b**) ours.

**Figure 3 sensors-23-06340-f003:**
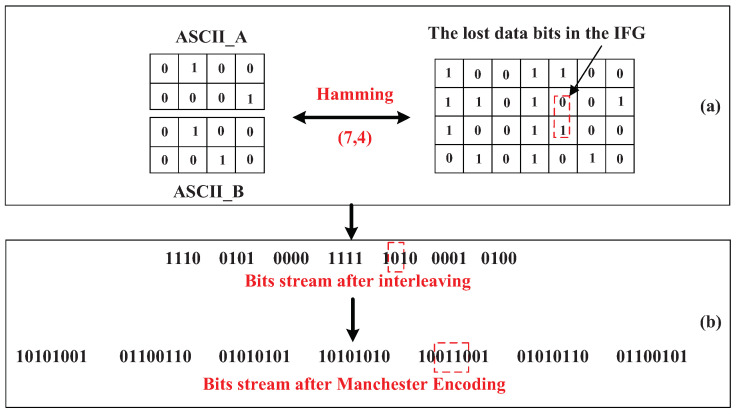
We show the system communication packets. (**a**) Linear block coding rule to encode transmitted data and (**b**) Bit interleaving and Manchester encoding for the low-speed visible-light-based services.

**Figure 4 sensors-23-06340-f004:**
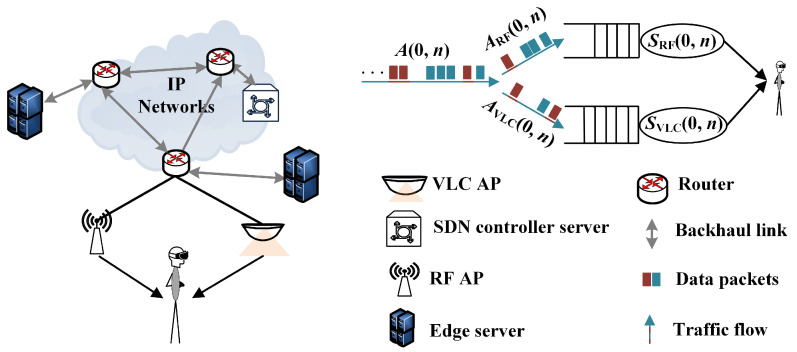
Scenario diagram for high-speed data services.

**Figure 5 sensors-23-06340-f005:**
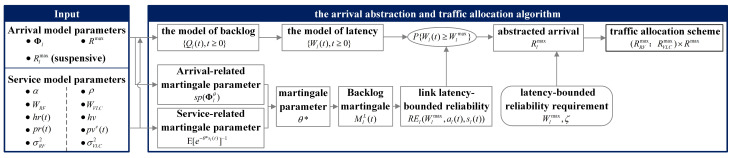
The summary of the arrival abstraction and traffic allocation.

**Figure 6 sensors-23-06340-f006:**
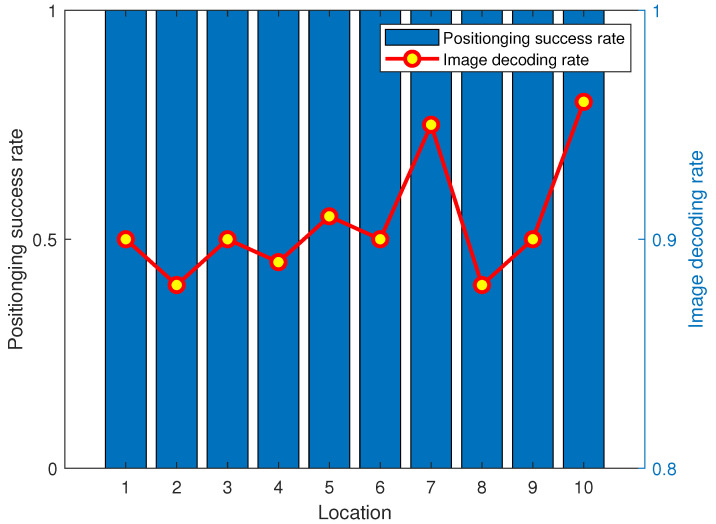
Image decoding rate and positioning success rate.

**Figure 7 sensors-23-06340-f007:**
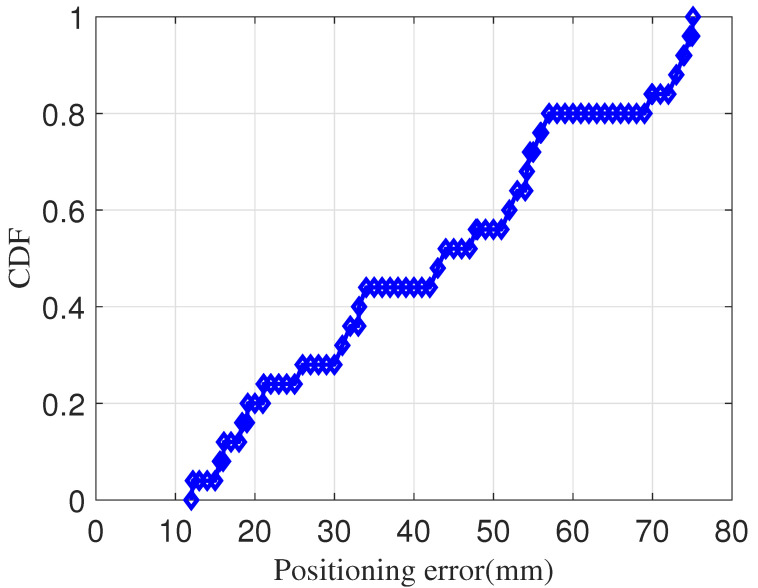
Positioning accuracy of the Fasys system.

**Figure 8 sensors-23-06340-f008:**
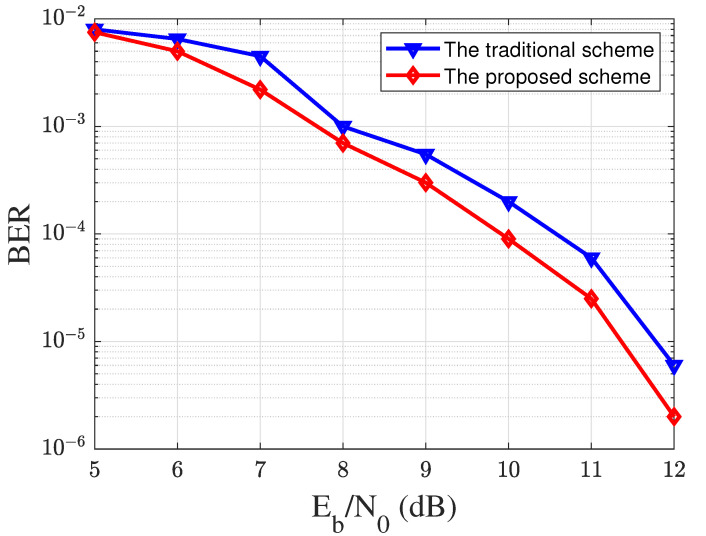
The system low-speed communication performance under different Eb/No conditions.

**Figure 9 sensors-23-06340-f009:**
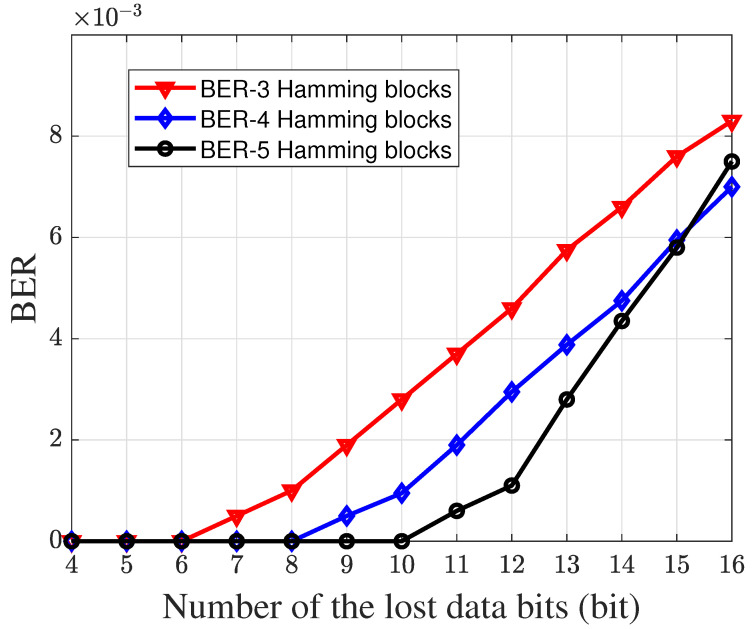
The system low-speed communication performance under different Hamming blocks conditions.

**Figure 10 sensors-23-06340-f010:**
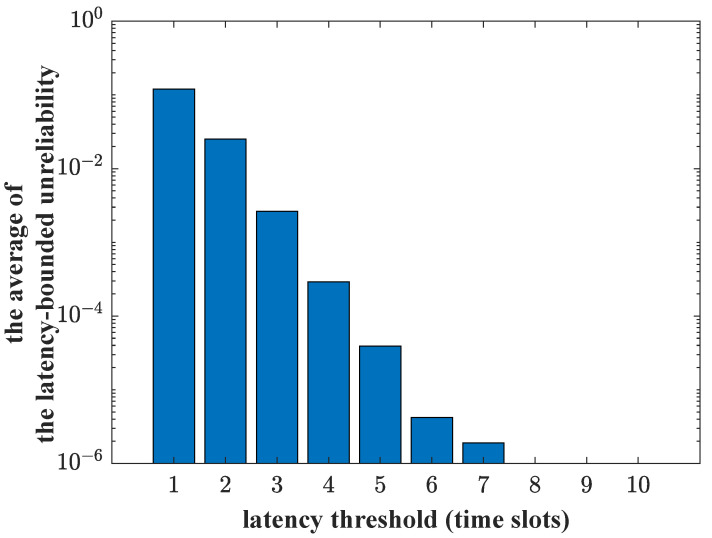
The average latency-bounded unreliability vs. latency threshold.

**Figure 11 sensors-23-06340-f011:**
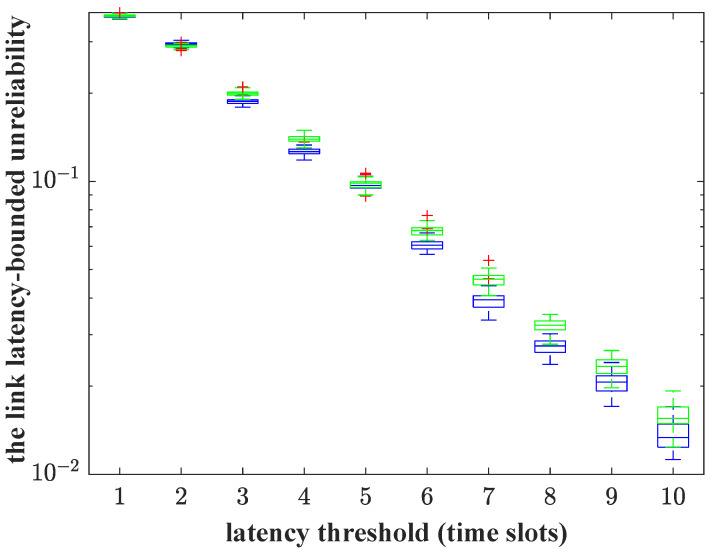
The latency-bounded unreliability vs. latency threshold. The green box-plots are the results that the RF link is only used and the blue box-plots are the results that the VLC link is only used.

**Figure 12 sensors-23-06340-f012:**
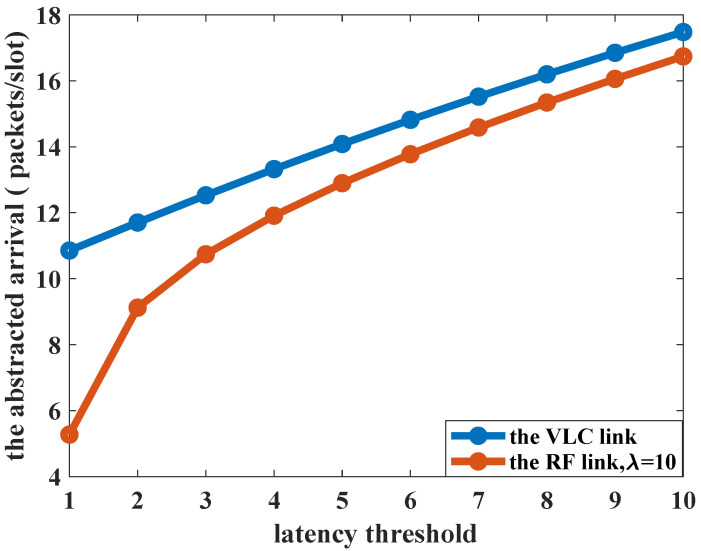
The abstracted arrival vs. latency threshold.

**Table 1 sensors-23-06340-t001:** The LED’s mapping mechanism for the positioning and low-speed data communication application.

LED transmission frequency/(bit/s)	$\frac{1}{290}\times 10^{6}$	$\frac{1}{285}\times 10^{6}$	$\frac{1}{280}\times 10^{6}$	$\frac{1}{275}\times 10^{6}$	⋯
The width range of the captured sequence header/pixel	$\left[128,130\right]$	$\left[126,127\right]$	$\left[123,125\right]$	$\left[121,122\right]$	⋯
Data carrying capacity per frame/bit	$\left[49,50\right]$	$\left[50,51\right]$	$\left[51,52\right]$	$\left[52,53\right]$	⋯

**Table 2 sensors-23-06340-t002:** The commonly used parameters for traffic allocation scheme.

	Parameters	Values
	The transition probability of the MMOO model, *p*	0.1
	The transition probability of the MMOO model, *q*	0.5
The traffic model	The peak rate of the MMOO model, $R^{max}$	24 packets/slot
	The length of a packet	512 bits
	The unreliability requirement, ζ	$10^{-4}$
	The bandwidth of an RB, $W_{RF}$	0.1 MHz
	The length of a time slot, *T*	0.5 ms
The RF link	The noise power spectral density in the RF network, ${\sigma_{RF}}^{2}$	−76 dBm/Hz
	The transmission power of the RF AP, $P_{r}$	2 W
	The RB allocation factor, λ	10
	The bandwidth of the VLC network, $W_{VLC}$	0.4 MHz
	The maximum electronic transmission power of the VLC AP *i*, $P_{v}^{e}$	20 W
	The noise power spectral density in the VLC network, ${\sigma_{VLC}}^{2}$	$10^{-21}$$A^{2}$/Hz
	The half-intensity radiation angle, $\vartheta_{1/2}$	70 deg.
The VLC link	The area of the PD, $A_{p}$	$10^{-4}$ m${}^{2}$
	The gain of the optical filter, $g_{of}$	1
	The refrative factor, *r*	1.5
	The semi-angle of FoV of the PD, Θ	50 deg.
	The responsivity of the PD, π	1
	The O/E conversion factor, ρ	1
The simulations	The number of observation slots	$10^{5}$
	The number of repetitions of the simulation experiments	50

## Data Availability

Not applicable.

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
