# Peer review of "Fasys: Visible-Light-Based Communication and Positioning Services towards Smart Cities"

_sensors, 2023, doi:10.3390/s23146340_

Round 1

Reviewer 1 Report

To push the VLC into realistic hybrid applications for supporting low-speed/high-speed communication and positioning scenarios, this submission proposes Fasys as a smart VLC-based fusion application system. In particular,  for low-speed services, a novel linear block coding and bit interleaving mechanism were used to enhance the accuracy of positioning and communication. For high-speed services, a statistical reliability analysis framework was proposed, which satisfies the latency and reliability requirements of high-speed applications. Extensive experiments show the superiority of the novel methods.  Generally, it is a very interesting work and holds its contribution for accelerating to accommodate real applications. However,the writing should be significantly improved.

1. On line 94 of page 3, “100% bit error rate” is not correct.

2. On line 398 of page 4, the correspondence among different packet formats is not correct.

3. On line 179 of page 5, the first occurrence of the abbreviation “RB” requires a full explanation.

4. On line 179 of page 6, what is the meaning of “Rl(t)”?

5. On line 209 of page 7, the first occurrence of the abbreviation “SNC”requires the full explanation.

6. In Figure 8, only the BER result can not prove the system can achieve a higher communication rate.

7. The bit interleaving was thought to spread possible errors to different blocks, there are no experiments that can prove the effectiveness of the bit interleaving mechanism.

Reviewer 2 Report

This paper proposed a smart visible light-based fusion application system which is aimed to solve problems that visible light-based system cannot achieve high-speed communication, low-speed communication, and positioning simultaneously and avoid problems caused by big data traffic with heterogeneity.

However, analyses and discussions are insufficient in this manuscript. Reconsideration is required after the authors addressing the following concerns. I expect that you could answer these comments carefully before publication.

1.In line 50, you addressed that (1) For indoor low-speed data services, when the visible light-based system tries to achieve positioning and communication simultaneously, the system is difficult to extract distinguishable positioning landmark features without affecting communication performance. It could be better to provide evidences or references to prove your opinions. Same questions appear in line 55 too, references should be given properly to verify your statements.

2.In line 67, you mentioned that “we find a robust linear mapping relationship between the data transmission rate and the width of the bright and dark stripes in the LED contour sub-maps”. Could you please provide several figures like the LED contour sub-maps under different transmission rate to show the linear mapping relationship you mentioned? If there are any references have the same point of view, it’s better to provided here.

3.In line 93, you concluded that “our Fasys system can achieve centimeter-level positioning accuracy and Kbps communication rate under 100 %-bit error rate (BER) for the low-speed data services”, could you please double-check the condition “100% BER”? 100% BER means there is no correct bit.

4. In figure.6, could you please explain what the “communication success rate” is? Besides, all 100 % results showed that the system has good performance but the responsivity of this system should be given.

Moderate editing of English language required

Reviewer 3 Report

In this paper authors propose a visible light-based fusion application system, named Fasys, which can achieve high-speed, low-speed communication, and high-precision positioning services.

I have the following concerns regarding this manuscript.

a) My main concern is the quality of the presentation. A high level architecture of the proposed system configuration can be included in a revised version and its components should be analyzed in more detail.

b) The queuing analysis should be better presented and further motivated. In its current form seems somewhat difficult to follow. 

c) How BER results has been obained? More detailed analysis is required.

Ν/Α

Round 2

Reviewer 1 Report

Thanks for your revision, this reviewer has no further comments.

Reviewer 3 Report

I am satisfied with the current version of this manuscript.

N/A